# Heat Shock Proteins and PD-1/PD-L1 as Potential Therapeutic Targets in Myeloproliferative Neoplasms

**DOI:** 10.3390/cancers12092592

**Published:** 2020-09-11

**Authors:** Steven De Almeida, Mathilde Regimbeau, Gaëtan Jego, Carmen Garrido, François Girodon, François Hermetet

**Affiliations:** 1HSP-Pathies Team, INSERM, LNC UMR1231, University of Bourgogne Franche-Comté, F-21000 Dijon, France; dealmeida.steven.d@gmail.com (S.D.A.); mathilde.regimbeau@gmail.com (M.R.); gaetan.jego@u-bourgogne.fr (G.J.); cgarrido@u-bourgogne.fr (C.G.); 2UFR des Sciences de Santé, University of Bourgogne Franche-Comté, F-21000 Dijon, France; 3Centre Georges François Leclerc, F-21000 Dijon, France; 4Haematology laboratory, Dijon University Hospital, F-21000 Dijon, France

**Keywords:** heat shock proteins, myeloproliferative neoplasm, PD-1/PD-L1, targeted therapy

## Abstract

**Simple Summary:**

Myeloproliferative neoplasms (MPN), which are a heterogeneous group of rare disorders that affect blood cell production in bone marrow, present many significant challenges for clinicians. Though considerable progress has been made, in particular with the JAK1/2 inhibitor ruxolitinib, more effective alternative therapeutic approaches are needed. In the search for new and more efficient therapies, heat shock proteins, also known as stress proteins, and the programmed cell death 1 (PD-1)/programmed death ligand 1 (PD-L1) immune checkpoint axis have been found to be of great interest in hematologic malignancies. Here, we review the therapeutic potential of stress protein inhibitors in the management of patients diagnosed with MPN and summarize the accumulating evidence of the role of the PD-1/PD-L1 axis in MPN in order to provide perspectives on future therapeutic opportunities relative to the inhibition of these targets.

**Abstract:**

Myeloproliferative neoplasms (MPN) are a group of clonal disorders that affect hematopoietic stem/progenitor cells. These disorders are often caused by oncogenic driver mutations associated with persistent Janus kinase (JAK)/signal transducer and activator of transcription (STAT) signaling. While JAK inhibitors, such as ruxolitinib, reduce MPN-related symptoms in myelofibrosis, they do not influence the underlying cause of the disease and are not curative. Due to these limitations, there is a need for alternative therapeutic strategies and targets. Heat shock proteins (HSPs) are cytoprotective stress-response chaperones involved in protein homeostasis and in many critical pathways, including inflammation. Over the last decade, several research teams have unraveled the mechanistic connection between STAT signaling and several HSPs, showing that HSPs are potential therapeutic targets for MPN. These HSPs include HSP70, HSP90 (chaperoning JAK2) and both HSP110 and HSP27, which are key factors modulating STAT3 phosphorylation status. Like the HSPs, the PD-1/PD-L1 signaling pathway has been widely studied in cancer, but the importance of PD-L1-mediated immune escape in MPN was only recently reported. In this review, we summarize the role of HSPs and PD-1/PD-L1 signaling, the modalities of their experimental blockade, and the effect in MPN. Finally, we discuss the potential of these emerging targeted approaches in MPN therapy.

## 1. Introduction

Myeloproliferative neoplasms (MPN) are clonal disorders in which stem/progenitor hematopoietic cells exhibit excessive cell proliferation for one or multiple myeloid blood lineages. In these disorders, the JAK/STAT signaling pathway is one of the most frequently mutated signaling pathways, notably with the *JAK2V617F* mutation forming a constitutively active mutant of JAK2 [1]. Therefore, the inhibition of JAK signaling has been envisioned as a therapeutic strategy [2,3,4]. The JAK1/JAK2 inhibitor ruxolitinib received approval for use in primary myelofibrosis (MF), and then for polycythemia vera (PV), improving disease-related symptoms but not preventing MF or progression to acute myeloid leukemia [5]. Since this current treatment is unable to stop or reverse these diseases, novel treatment options are being actively sought in order to achieve curative outcomes. Recent studies have investigated alternative therapeutic strategies targeting other signaling pathways or proteins in combination with JAK1/JAK2 inhibition. These studies have looked at PI3K/AKT/mTOR [6,7], MAPK signalization [8] or Bromodomain and extra-terminal domain (BET) proteins [9], which demonstrated significant and promising synergism in various in vitro models or in conditional *Jak2V617F* knock-in mice. In this search for new and more efficient therapies [10], heat shock proteins (HSPs) and, more recently, the programmed cell death 1 (PD-1)/programmed death ligand 1 (PD-L1) axis have been found to be of great interest in hematologic malignancies. It is thought that these targets may become part of the therapeutic arsenal against MPN.

Since HSPs, such as HSP90, HSP70, and HSP27, are major proteins of the neoplastic cell, targeting these proteins offer interesting perspectives for the treatment of hematologic malignancies [11,12]. HSP90 and 70 have received the most interest, and the development of HSP90 inhibitors is by far the most advanced. Currently, approximately 20 of these inhibitors have undergone clinical evaluation [13,14]. As outlined below, HSP inhibition in MPN has provided promising results over the past few years, and several clinical trials evaluating the safety and efficacy of HSP90 inhibitors (PU-H71; NCT01393509 and NCT03935555) have recently been initiated (source: https://clinicaltrials.gov). Research into cancer immunotherapy through the blockade of immunological checkpoints, which earned the Nobel Prize in Physiology or Medicine in 2018 [15,16], has made considerable progress. As a result, novel anticancer therapies have been tested in numerous solid neoplasms including melanoma, non-small cell lung carcinoma (NSCLC), renal cell carcinoma (RCC) [17], and more recently in hematologic malignancies including leukemia [18], in which there are ongoing clinical trials for nivolumab (NCT02011945, NCT02275533), pidilizumab (NCT01096602), and pembrolizumab (NCT02332980) (source: https://clinicaltrials.gov).

The purpose of the present review is to focus on the therapeutic potential of HSP inhibitors in the management of patients diagnosed with MPN and to summarize the accumulating evidence of the role of the PD-1/PD-L1 axis in MPN, providing perspective on future opportunities relative to the inhibition of these targets.

## 2. Heat Shock Protein Chaperones

HSPs, also known as stress proteins, are highly conserved molecular chaperones that are a part of the protein quality control machinery that helps cells survive stressful conditions [12]. HSPs have been classified according to their molecular weight: there are four main families of high molecular weight proteins (HSP110, HSP90, HSP70, and HSP60) and one of low molecular weight proteins (small HSPs). The expression of HSPs can be constitutive or induced by physiological or environmental stress. Stress conditions include oxidative stress, hypoxia, inflammation, and chemotherapeutic agents [19]. In physiological conditions, HSPs are needed to ensure essential cellular roles by means of their chaperone function: they contribute to the correct folding of neosynthesized or denatured proteins, thereby guaranteeing their functional structure. They prevent the aggregation of misfolded proteins, assure the transport of proteins through cellular compartments, and even participate in certain signaling pathways [20]. Otherwise, under stress conditions, HSPs allow cells to survive through mechanisms such as the inhibition of key effectors of the apoptotic machinery, and their “protein triage” function, which leads misfolded proteins to proteasome-mediated degradation [21,22,23]. Some HSPs (HSP27, HSP70, HSP90, and HSP110) are secreted under two different forms: free soluble or within small vesicles [24,25,26,27] through which the secreted HSPs are responsible for certain types of cell signaling through membrane receptors and/or immune-cell dysregulation [28,29]. Due to their intracellular and extracellular functions, high expression levels of HSPs are associated with poor prognosis and treatment resistance in many diseases including a wide range of cancers. They are also associated with certain hallmarks of cancer, such as cell proliferation, invasion and metastasis [30]. Moreover, HSPs play a role in the development of MPN [31,32]. Here, we outline the current understanding of the role of HSPs, including HSP27, HSP70, and HSP90, in MPN, and summarize the recent advances in utilization of HSP inhibitors as emerging targeted therapies for MPN.

### 2.1. HSPs and Their Targeting in MPN

The strong relationship between the JAK/STAT signaling pathway and HSPs was recently reviewed by Jego et al. [33]. The authors showed that most HSPs can be regulated by STAT proteins both in stress and non stress situations such as heat shock response [34], adaptation to reactive oxygen species [35], cell growth, and in response to many other stimuli [36,37]. Conversely, HSPs control STAT3/STAT5 activation in cancer [33]. Based on these observations, it has become obvious that HSPs are of potential interest in MPN.

#### 2.1.1. HSP90 and Their Targeting in MPN

The human HSP90 family (hereinafter referred as HSP90) includes five members called HSPC1 to HSPC5, best known as HSP90, HSP90α, HSP90β, 94-kilodaltons (kDa) glucose-regulated protein (GRP94), and tumor necrosis factor receptor-associated protein 1 (TRAP1), respectively [38]. Structurally, these proteins are composed of an N-terminal domain which binds ATP and a C-terminal domain responsible for protein dimerization and client protein binding. They have various subcellular localizations, but they are mainly found in the cytoplasm, the endoplasmic reticulum, or the mitochondria [38]. HSP90 operates in a molecular complex involving co-chaperones (e.g., Hop1, PP5, P23, Cyp40, Aha, Tah1, Pih1, Cdc37, TTC4, Sgt1) and client proteins. HSP90 is involved in various processes in physiological conditions, such as protein folding, cell cycle or cell transport, and under stress conditions involving endoplasmic reticulum and oxidative stresses [39].

One of the key articles on HSPs in MPN was published by Marubayashi et al. [40]. In response to PU-H71 treatment (an HSP90 inhibitor), they observed a dose-dependent decrease in cell growth associated with JAK2 degradation in JAK2V617F cell lines such as UKE-1 and SET-2, in JAK2 and thrombopoietin receptor (MPL or EPOR) mutant transduced cells (JAK2K539L; hMPLW515L), and also in primary MPN patient samples (JAK2V617F PV). It was shown by immunoprecipitation that JAK2 is a client protein of HSP90, and the PU-H71 treatment of UKE-1 cells reduced the STAT5a transcriptional program. Furthermore, this inhibitor reduced the mutant allele burden in Jak2V617F mice, which led to a decrease in white blood cells, hematocrit, and Ter-119 expression in bone marrow. At the same time, the authors reported a decrease in white blood cells, platelets, and spleen megakaryocyte in the MplW515L mutant mice model [40]. The growing evidence regarding HSP90 interactions with JAK/STAT signaling, both in vitro [41] and in vivo [40], supports the idea that HSP90 inhibition might be of therapeutic interest in JAK/STAT dysregulated diseases such as MPN [42]. Interestingly, PU-H71 monotherapy improved the survival of Jak2V617F-engrafted Tp53-KO leukemic mice and was associated with a strong decrease in splenomegaly [43]. The combination of ruxolitinib and PU-H71 was also assessed in mice, resulting in a decrease in JAK2 signaling activity [44].

Fiskus et al. demonstrated the efficacy of NVP_AUY922 (AUY922), another HSP90 inhibitor, on HEL 92.1.7 and UKE-1 cells [45]. They noted a decrease in the abnormal cell cycle, with an accumulation of cells in G2/M and increased apoptosis. Similar to the work of Marubayashi et al. with PU-H71, AUY-922 treatment also depleted JAK2 and its downstream signaling in cells overexpressing *JAK2V617F*, and, once again, JAK2 was identified as an HSP90 client protein. The relevance of co-treatment with a JAK2 inhibitor (TG101209) and an HSP90 inhibitor (AUY-922) has been made on MPN cell lines and primary CD34+ MF patient cells. This combination of inhibitors had a synergistic effect on apoptosis and activation of the JAK/STAT signaling pathway [45]. Moreover, three JAK inhibitor-resistant mutants were identified in B-cell acute lymphoblastic leukemia, and in vivo results demonstrated that HSP90 inhibition by AUY922 and 17-AAG targeted JAK2 and overcame resistance to enzymatic kinase inhibitors including BVB808, a novel JAK2 inhibitor of the N-aryl-pyrrolopyrimidine scaffold class [46]. These results are consistent with another in vitro study published in 2012 [47].

An HSP90 inhibition-based therapeutic strategy was, therefore, envisaged for use in MPN, especially in response to JAK-inhibitor-resistant mutations [48,49]. Following a phase 1 clinical trial using alvespimycin in acute myeloid leukemia patients [50] and a phase 2 study on NSCLC with IPI-504 [51], it was also thought that HSP90 inhibitors could be fairly well tolerated. Unfortunately, a 2018 phase 2 clinical study showed that, despite the efficiency of the dual inhibition of JAK2 and HSP90, the HSP90 inhibitor AUY922 induced severe adverse effects including decreased visual acuity, night blindness and gastrointestinal bleeding [52]. 

Finally, PU-H71 has not been assessed in MPN though three clinical trials focusing on MF are in progress (NCT01393509, NCT03935555 and NCT03373877; source: https://clinicaltrials.gov). Unfortunately, it is not known if MPN patients will tolerate the drug, but a clinical trial on patients with solid tumors was encouraging. While this study was aborted prematurely due to discontinuation of the drug supply, the patients treated with 2–6 cycles of 10–470 mg/m²/day showed grade 2–3 adverse effects but did not reach any dose limiting toxicities. The best response of stable disease was observed for 35% of the population (six patients) [53] (NCT01581541; source: https://clinicaltrials.gov).

#### 2.1.2. HSP70 and Their Targeting in MPN

The HSP70 family is a large family of highly conserved proteins of about 70 kDa. Similar to HSP90, HSP70 are composed of two main domains: a nucleotide-binding domain which hydrolyzes ATP in the N-terminal region [54], and a C-terminal substrate-binding domain [55]. HSP70 has been widely studied, and the published data show that this family of proteins is involved in a considerable number of signaling pathways, including stress response, apoptosis, cell growth, and immunomodulatory functions [56]. 

In MPN, the major findings regarding HSP70 highlight its role in erythropoiesis. In the past, our laboratory showed that HSP70 is required for normal erythropoiesis because it protects GATA-1 from caspase-3 cleavage in the nucleus during the terminal differentiation of erythroid progenitors [57]. Following this discovery, it was shown that the cytosolic-nuclear shuttling of HSP70 was the critical event in this mechanism [58]. In beta-thalassemia, HSP70 sequestration by the free alpha globin chains is a major cause of ineffective erythropoiesis because it prevents nuclear translocation. However, this translocation is restored by the transduction of a nuclear form of HSP70 or a caspase-3-uncleavable GATA-1 [59]. 

Other researchers have also shown the importance of HSP70 in erythroid lineage proliferation and survival in cell lines and MPN patient samples [60]. A synergic effect was shown when the HSP70 inhibitor KNK437 was used in vitro in both PV patient cells and in a *JAK2V617F* cell line in association with the well-known JAK inhibitor ruxolitinib [61]. Overall, HSP70 has been correlated with a poor prognosis in myelodysplastic syndromes [62], and, although the implication of HSP70 in MPN needs to be further investigated, targeting HSP70 could be a promising therapeutic strategy. 

#### 2.1.3. HSP27 and Its Targeting in MPN

Among the small HSP family, HSP27 is a low molecular weight ATP-independent HSP [19] that is strongly induced by proteotoxic stresses, such as anticancer drugs, oxidative stress, radiation, or inflammatory shock, as well as physiological processes like cell differentiation [31]. Two forms of HSP27 have been identified depending on their phosphorylation and oligomerization status, and each form is associated with a different function and subcellular localization. Whereas HSP27 phosphorylated monomer is involved in cytoplasmic and nuclear signaling networks, the multimeric (non-phosphorylated) form is mainly involved in protein folding and anti-aggregation activity [63]. Upregulation of HSP27 expression and phosphorylation has been associated with numerous aspects of neoplastic disease, including disease features, metastatic behavior, resistance to treatment, and poor prognosis [64]. It has been demonstrated that HSP27 is needed for erythroid differentiation and that it is involved in late erythropoiesis by regulating GATA-1 levels through ubiquitination [65,66]. Recently, our team described the role of HSP27 in MF and its impact on STAT5 de-phosphorylation and, therefore, its potential therapeutic interest in MPN [67]. Using two mouse models of MPN-related MF (TPO^high^ and *Jak2V617F*), we showed that HSP27-specific inhibition by means of an antisense oligonucleotide, or OGX-427 (Apartosen), limits overall disease progression (reduced spleen weight and size, decreased myeloid proliferation and megakaryocytic expansion both in spleen and bone marrow). Our team also demonstrated for the first time the interaction between HSP27, JAK2, and STAT5, and the rise of HSP27 in patient samples, supporting the chaperoning role and the implication of HSP27 in MPN pathogenesis. Altogether, these findings highlight the potential interest of HSP27 inhibitors as a complementary therapy for MPN. Nevertheless, new strategies for HSP27 inhibition using molecules that are more druggable and more potent but just as selective as OGX-427 need to be promoted in the future.

The known HSP roles and inhibitors in MPN are synthetized in Figure 1, and the main strategies for HSP inhibition in MPN are summarized in Table 1.

## 3. Recent Insights for HSP Inhibition in Cancer

Several other HSP inhibition modalities have recently been developed and proposed as alternative cancer therapies; some have entered clinical trials and may become viable options for treating MPN in the future. It is undeniable that HSP90 plays a major role in MPN, and a growing number of publications have proven its involvement in pathogenesis and treatment resistance. However, the first clinical trial that attempted to inhibit HSP90 was a disappointment: patients developed serious adverse effects, and the relevance of HSP90 inhibition as a therapeutic strategy in MPN was called into question [52]. Nevertheless, HSP90 inhibitors could still be improved in order to obtain better efficacy or diminish adverse effects. Indeed, the therapeutic potential of HSPs led to the development of a number of HSP inhibitors for cancer, most of which have been recently reviewed by Jego et al. [10]. Since then, new inhibitors have also been developed, and some have been tested in clinical trials.

### 3.1. HSP90 Inhibitors

In one phase 2 clinical trial, the HSP90 inhibitor ganetespib was used in a subset of patients who developed esophagogastric cancers with specific genetic abnormalities. Despite the manageable toxicity of ganetespib, this clinical trial was halted because only a subset of patients showed a minor response, and the single-agent activity was deemed insufficiently efficient [68].

Since then, many compounds have been identified as HSP90 inhibitors in vitro, such as vibsanin A analog C [69] or C086 [70]. Among these new molecules, KW-2478, a novel HSP90 inhibitor, was shown to induce a downregulation in the expression of STAT3 [71]. Thus, it might be interesting to test this molecule in MPN models. Seeing as HSP90 is a family of proteins containing multiple proteins, Jung et al. also developed paralog-specific inhibitors to reduce toxicity and increase drug specificity. This approach used three synthesized compounds called 18, 19, and 30, which are 350-fold more specific to HSP90 than TRAP1, and from one- to three-fold more specific than GRP94 [72].

Finally, other strategies based on the development of dual inhibitors are ongoing. For example, DHP1808 is a dual inhibitor inhibiting both PI3K and HSP90 proteins [73], and, in the same manner, “compound 4” was designed from hybridization for both PARP and HSP90 inhibition [74]. This approach, based on the development of dual inhibitors, could be adapted to MPN models to reduce the adverse effect of HSP90 inhibitors and increase treatment efficacy.

In breast cancer models, new HSP90 inhibitors have been tested in vivo for their anti-tumoral properties since 2019. BJ-B11 was assessed for tumor growth inhibition and showed satisfactory results in mice [75]. The same observations were made for AT-533, another well-known HSP90 inhibitor, with a decrease in angiogenesis and more efficacy than 17-AAG [76]. Among the new HSP90 inhibitors, DN401 is a novel mitochondrial permeable pan-HSP90 inhibitor simultaneously targeting HSP90, GRP94, and TRAP1 DN401 [77]. Finally, TAS-116 is an HSP90-specific inhibitor that has shown encouraging results, with low toxicity in imatinib-mesylate-resistant gastrointestinal stromal tumors in mice, as well as in lung cancer cell lines [78]. 

These recent observations suggest that the new HSP90 inhibitors might be worth testing in the field of hematologic research. 

### 3.2. HSP70 Inhibitors

If the development of more specific HSP90 inhibitors can abrogate the adverse effects observed in the first clinical trials, then HSP90 inhibition may be promising for MPN. Yet, in cancer, it has been observed that HSP90 inhibition often leads to a compensation through HSP70 expression [79]. If this phenomenon also occurs in MPN, blocking both HSP70 and HSP90 may yield interesting results. Fortunately, there has been much progress in the development of HSP70 inhibitors. JG2-98 is a derivate of JG-38, which is derived from the MKT-077 compound [80]. In vitro, these compounds have been shown to exhibit anti-proliferative effects in breast and prostate cancer [81]. In triple negative breast cancer MDA-MB-231 xenograft mouse models, these molecules led to a reduction in tumor burden, which was attributed to the inhibition of the HSP70-BAG interaction [82]. Interestingly, the level of the well-known JAK2 downstream protein AKT was also strongly decreased upon treatment. The authors identified other compounds derived from JG-38 (JG-231 and JG-294) with a mid-nanomolar potency that were described as less toxic than JG-38 for non-tumor cells. Finally, peptide aptamers are another type of molecule developed to inhibit HSP70 in cancer therapy. These molecules, which are used as an alternative to neutralizing antibodies, target different regions of HSP70. They were found to block tumor progression by inducing apoptosis and to create an anticancer response involving a strong intratumor infiltration with CD8 cells and macrophages [83,84].

## 4. PD-1/PD-L1 Axis and Its Targeting in Cancer

Over the past decade, multiple immunotherapy approaches have emerged in cancer treatment [85]. These advances rely on our growing understanding of tumor immune escape mechanisms and discoveries regarding immune checkpoint inhibitors and immune co-stimulating molecules, such as the PD-1 receptor and its ligands PD-L1 and PD-L2 [86]. 

### 4.1. Overview of the PD-1/PD-L1 Checkpoint Pathway

First described as a “two-signal” activation theory allowing the self to be distinguished from the non-self [87], it was only later that the involvement of negative co-stimulatory signals was found to exist. What we now call “immune checkpoints” play a crucial role in maintaining peripheral tolerance and hampering autoimmunity [88,89]. Conversely, immune checkpoint inhibition favors T cell activation, which is critical for enhancing antitumor responses. Since they were discovered, co-inhibitory molecules have been obvious therapeutic targets because they promote T cell activation, which is needed to enhance antitumor response. The objective is, therefore, to remove the co-inhibitory signals that block T cell responses rather than to re-activate immune cells. 

#### 4.1.1. Discovery of the PD-1/PD-L1 Pathway

PD-1, which was first identified as an apoptotic molecule [90], is a surface protein mostly expressed on activated T cells [91]. Many aspects of the PD-1 pathway have been studied since 2000, including their ability to impair the repertoire of mature T cells and their contribution to autoimmune diseases [88,91]. Several teams worked simultaneously to discover the PD-1 ligand (PD-L1) [92] and to define this axis as a negative regulator of immune response.

#### 4.1.2. Expression and Distribution of PD-1 and Its Ligands

PD-1 is expressed in a large range of cells including activated T cells, monocytes, natural killer (NK) cells, dendritic cells (DC), myeloid cells, and antigen presenting cells (APC) [93,94,95,96,97]. While PD-L2 is mostly expressed by activated macrophages and dendritic cells, PD-L1 is carried by T- and B-lymphocytes, monocytes and macrophages [95], which gives this pathway broader potential for immune modulation on a number of levels [98]. Moreover, PD-L1 expression is induced by many cytokines [89,92,99], and intracellular signal transducers such as myeloid differentiation factor 88 (MyD88) or JAK2 are acknowledged as playing a major role in signaling pathways involved in PD-L1 expression [100,101]. Finally, PD-L1 is also expressed by tumor cells of various origins, including hematologic. Immune cell inhibitory signals are triggered after ligation with PD-1 at the surface of these tumor cells, causing antitumor immunity to become ineffective, and allowing the tumor cells go undetected by the immune system [102,103].

#### 4.1.3. Effect of PD-1/PD-L1 Interaction 

The interaction between PD-1 and PD-L1 turns off the ability of cytotoxic T cells to be activated or to proliferate [104,105], creates cell exhaustion and metabolic alterations [106,107], and affects cell cycle activity [108]. As described by He and Xu and Nurieva et al., PD-1 signaling impairs T cell activation and activity through its effects on multiple transcription factors such as GATA-3 or T-bet [109], and multiple signaling pathways including PI3K/AKT and RAS/MEK/ERK [110]. Finally, considering the large scope of its actions, PD-1 signaling has been defined as a key target in immunotherapy seeing as successful inhibition of this axis would help restore T cell function. 

### 4.2. Effects of Signaling Pathways on PD-1/PD-L1 and Basis of Anti-PD-1/PD-L1 Therapies in Cancer

While the PD-1/PD-L1 axis is considered essential for physiological immune response modulation and autoimmunity, numerous studies have demonstrated that tumors exploit PD-1-mediated immune properties to escape physiological immune response. Thus, oncogenic mutations or inflammation mainly mediate PD-L1 expression by cancer cells [111,112], finally leading to tumor resistance [113] and promotion of tumor growth by means of antiapoptotic signals [114]. A wide range of solid tumors and hematologic malignancies, including MPN, have been found to overexpress PD-L1, strongly suggesting that this axis is of therapeutic interest. Though PD-L1 expression correlates with intensified tumor growth and invasiveness, which is associated with adverse prognosis [115,116,117], the presence of PD-L1 within the tumor microenvironment predicts a better clinical response to PD-1/PD-L1 checkpoint blockade therapy [17,118]. As a result, the PD-1/PD-L1 pathway has been associated with multiple roles in carcinogenesis, from promoting tumor growth, proliferation, and survival, to additional features such as metastasis, chemoresistance, and poor prognosis [119]. That is why blockade therapies targeting the PD-1/PD-L1 pathway with anti-PD-1 or anti-PD-L1 antibodies aim to suppress cancer cell survival and promote T effector cell generation, contributing to enhanced antitumor T cell responses and tumor regression. 

### 4.3. History and Examples of PD-1/PD-L1 Inhibitors

The first immune checkpoint blocking strategy emerged in 2010 with the anti-CTLA-4 antibody ipilimumab, which was prescribed to melanoma patients [120]. Research on other co-inhibitory receptors such as PD-1 led to the development of new antibodies targeting PD-1/PD-L1 axis in melanoma. Notably, nivolumab and pembrolizumab delivered very good results in melanoma [121] before proving their efficiency in other cancers including RCC and NSCLC [17]. The repertoire of anti-PD-1 and anti-PD-L1 antibodies has continued to grow, and many formulas, with various affinities, have emerged. These include durvalumab, atezolizumab, and MDX-1105, which are currently under investigation for their use in a wide range of PD-1-sensitive tumors. 

Overall, no less than 500 clinical studies have been conducted worldwide for this type of antibody, and it has been used to treat solid cancers, such as melanoma, NSCLC, urothelial cancer, RCC [122], and colon, gastric and breast cancer [123]. If the potential of anti-PD-1 and anti-PD-L1 therapeutic agents is well documented in the context of solid tumors, these therapeutic approaches have not yet been fully applied in hematologic cancers. However, publications have emerged describing the involvement of the PD-1/PD-L1 in hematologic conditions and the potential benefit of PD-1 targeting strategies, particularly in MPN.

## 5. Targeting the PD-1/PD-L1 Axis in MPN

Most recent studies have attempted to demonstrate the role of the PD-1/PD-L1 axis in MPN, seeing as these molecules appear to be overexpressed by key cell populations. However, research still has many questions to answer before this therapy can be used in the context of routine MPN treatment.

### 5.1. PD-1/PD-L1 Expression in MPN

A recent study by Wang et al. demonstrated that significantly higher levels of PD-1 and PD-L1 were found in MPN patients (including ET, PV, and MF) compared with controls [124]. Moreover, still in the context of MPN, PD-L1 has been found to be expressed more abundantly on the membranes of megakaryocytes and myeloid cells, including monocytes [125], whereas PD-1 expression was found only on immune T cells. One study in 63 MPN patients found a rise in PD-1 and PD-L1 expression at the surface of many cell types including CD4+, CD8+, CD14+, and CD34+ cells, regardless of the patient’s mutational status [124]. Another study reported that more than 50% of PD-1 or PD-L1 expression was associated with enhanced response to PD-1 inhibition treatment in solid tumors, and a higher percentage of PD-1 and PD-L1 was found in circulating CD34+ cells in blood samples from MPN patients than in samples from healthy donors [126]. Nevertheless, when mutational status is taken into account, no link has been identified between JAK2, calreticulin (CALR), or MPL mutational burden and PD-1/PD-L1 expression in cells. However, Sorensen et al. demonstrated that treatment with the JAK2 inhibitor ruxolitinib, alone or combined with interferon-alpha (IFNa), lowered the high expression levels of PD-1 and PD-L1 in samples from MF and PV patients compared with healthy donors [127].

### 5.2. Regulation Mechanisms of PD-1/PD-L1 Expression in MPN

In a study by Prestipino et al., the JAK/STAT signaling pathway was found to interact with the mechanisms that regulate PD-L1 expression [128]. Using transfected mice models and patient-derived blood samples, this study demonstrated that *JAK2* mutation leads to an upregulation of PD-L1 promotor activity and STAT3-dependent PD-L1 expression in spleen and bone marrow cells. On the contrary, the inhibition of STAT3 or JAK2 proteins lowered PD-L1 expression. Constitutive activation of the *JAK2* oncogene led to the phosphorylation of STAT3 and STAT5 which then stimulated PD-L1 promoter activity and PD-L1 protein expression. The authors also showed that PD-L1 overexpression was found in various cell types, including T lymphocytes, monocytes, and myeloid-derived suppressor cells (MDSC), with a predominant impact on platelets and megakaryocytes (MK). Though monocytes and MK were the most affected by PD-L1 expression changes, these findings are consistent with data regarding the *JAK2* allele burden in MPN since they are the cells that are most affected by *JAK* mutations [129]. Prestipino et al. also showed that PD-1 inhibition treatment of *Jak2*-mutated MPN mouse models increased survival rates, decreased the *Jak2*-related allele burden, and favored CD8+ T cell effectiveness by re-establishing metabolism and cell cycle progression. Finally, they found that the cells carrying JAK2-mutated proteins showed enhanced PD-L1 expression, which intervened in T cell metabolism and inhibited cell cycle progression [128]. In another article, MPN patients were found to have increased MDSC levels [130]. This is consistent with the fact that MDSC interact with T cells, enhancing IL-10 production and STAT3 phosphorylation, which then leads to PD-L1 expression in those cells [131]. Increased TLR2 signaling has been found in patients with ET and PV [132], leading to MERK/ERK, STAT pathway activation and, finally, PD-L1 promotor activation. The regulation of PD-1 is not as well understood, but it seems to rely on persistent inflammatory stimulation of T cells, leading to a downregulation of some immune cell transcription factors such as T-BET or BLIMP-1, which causes immune T cell exhaustion and triggers PD-1 production [133]. These phenomena may explain the increase in PD-1 and PD-L1 in MPN patients. The role and implications of the PD-1/PD-L1 axis in MPN are summarized in Figure 2.

### 5.3. Immune Checkpoint Blockade in MPN

The relevance of PD-1/PD-L1 as a therapeutic target in MPN has now been described, but the question of the efficacy of blockade therapies remains and the mechanisms involved in the process also need to be clarified. As mentioned above, Wang et al. recently demonstrated that, in human patients, PD-L1 overexpression was correlated with enhanced response to treatment with anti-PD-1 antibodies. Moreover, Holmström et al. showed that vaccination with PD-L1-derived epitopes could improve the curative effects of blockade therapies [134]. They observed that 71% of patients displayed a significant immune response to PD-L1 in the early stages of MPN, and that CD4^+^ T cells were responsible for the immune response observed, hereby promoting anti-tumor immunity. 

When considering the mutational status of MPN, the *JAK2* mutations and in particular the *JAK2V617F* mutation should be taken into consideration. This mutation results in the generation of a mutated protein with high prevalence and a strong immunogenic potential, making it a key therapeutic target. In a recent study, Holmström et al. established a specific CD8^+^ T cell population that selectively recognizes the *JAK2V617F* mutation carried by MPN cells [135]. The *JAK2V617F* mutation can, therefore, be viewed as a tumor antigen and likened to a new treatment modality for MPN. 

Similar to previous work on *JAK2* mutations, an interesting study conducted by Bozkus et al. focused on blocking antibodies in the context of *CALR* mutations. CALR is a molecular chaperone whose gene has been found to be mutated in a significant proportion of MPN patients. It is worth noting that all patients carrying *CALR* mutations produce an altered protein containing a very characteristic C-terminus sequence [136]. Bozkus and colleagues demonstrated that mutated-CALR can be seen as a shared neoantigen, potentially eliciting antigen-specific response from both CD4^+^ and CD8^+^ T cells against MPN transformed cells. Notably, the presence of PD-1 abrogated responses, and the use of the monoclonal blocking antibody pembrolizumab restored mutant-CALR-specific T cell immunity. This discovery confirmed that mutant forms of CALR are a MPN-specific neoantigen, paving the way for the design of new immunotherapies targeting both the PD-1/PD-L1 axis and CALR mutants in MPN.

There is considerable evidence suggesting the interest of therapies targeting the PD-1/PD-L1 axis in MPN. However, it is clear that more research will be needed to understand the links between the various disease-related factors. Currently, phase II clinical trials are being conducted to test the efficacy of anti-PD-1 blocking antibodies in the context of MPN (NCT02421354, NCT03065400; source: https://clinicaltrials.gov).

## 6. Conclusions and Outlook

Though the HSP90 inhibition strategy was seen as a promising therapy for MPN, treated patients developed severe adverse effects, leading to disappointing results. This failure resulted in uncertainties regarding the feasibility of HSP90 inhibition: would it possible to abrogate the adverse effects by developing new, more specific inhibitors, or is it impossible to avoid adverse effects that are inherent to the numerous client proteins of this chaperone?

Growing evidence suggests that there is still hope that HSP inhibition can be used in MPN, especially seeing that other members of the HSP family, such as HSP27, have recently emerged as proteins of interest in MF. However, extensive research will be needed to promote HSP inhibition therapy in MPN. To date, only one publication is available for HSP27 [67], and HSP70 has been mostly investigated in the context of erythropoiesis [57,59]. 

Beyond the well-known canonical intracellular functions of HSPs, their extracellular presence and functions were only recently discovered. It has been reported that HSPs can be secreted through extracellular vesicles (EVs) and that high expression of HSPs in EVs is a negative prognostic factor in metastatic solid cancers [137,138]. Surprisingly, when compared to its intracellular form, extracellular HSP70 has been shown to have a contrasting effect on neuroprotection in certain neurologic disorders [139]. Finally, extracellular HSPs could also be of great interest for the diagnosis of diseases. Indeed, their high expression in EVs in conditions such as Alzheimer’s [140] and various types of cancer [27] makes them useful biomarkers. If further investigation into these proteins is conclusive, they could be used in future diagnostic approaches for MPN.

New information regarding the treatment of MPN has come to light for another signaling pathway, which is quite unexpected since the immune aspect of these diseases has been mostly overlooked. In addition, though the PD-1/PD-L1 axis has been a popular subject in cancer research, little has been done in MPN until very recently. Wang et al. and Prestipino et al. initiated the first studies on PD-1/PD-L1 in MPN, and their results show that this immune axis could be a relevant therapeutic target for MPN [124,128] because it seems to be triggered by key actors of the disease’s development (oncogenic driving mutations), and it is also involved in immune silencing. 

The future of cancer therapeutics will likely rely on combined methods seeing as drugs can have multiple targets and the administered doses are often low. While PD-1 blocking antibodies appear to be associated with positive effects on their own, it is reasonable to question whether the combination of this treatment with compounds that are already used in MPN could provide greater benefits to patients. In any case, the choice of combinations should be reasoned and relevant. 

The use of IFNα, a compound endowed with anti-proliferative properties [141,142], could potentiate the immuno-stimulating effects of antibodies directed against PD-1 and, thus, make it possible to modulate several aspects. 

Likewise, it would be relevant to test a combination of blocking antibodies and agents targeting neoantigens expressed by MPN cells. Indeed, recent studies have shown the existence of specific immunity directed against *JAK2V617F*- and *CALR*-mutated cells [135,136], as well as the existence of neoantigens, such as CD123, making it possible to discriminate pathological cell clones [143]. However, anti-tumor immune responses are silenced by overexpression of the PD-1/PD-L1 pathway. Thus, inhibitor combinations of PD-1/PD-L1 and JAK2 or even PD-1/PD-L1 and CD123 are worth considering and could prove beneficial for patients. 

While JAK2 inhibitors would reduce the inflammatory phenomena associated with MPN, anti-PD-1 immunostimulating antibodies would help restore an effective immune response. Additionally, the growing data on the role of CD123 and the promising results of clinical studies carried out with inhibitors of this molecule (in particular Tagraxofusp, a CD123-directed cytotoxin consisting of the fusion of interleukin-3 with a truncated diphtheria toxin payload) make it a target of choice in the treatment of MPN [143]. The promising results presented during EHA2020 suggest that Tagraxofusp has a clinical benefit with limited risk for MF patients (NCT02268253.6.7.0; source: https://library.ehaweb.org/eha/2020/eha25th/295038/). Likewise, the combined use of a CD123 inhibitor and blocking antibodies would make it possible to target the transformed cells and to reactivate the immune responses directed against the tumor cells at the same time.

It is now well known that two forms of PD-L1 are crucial in tumor immune escape in general. Many studies have highlighted the existence of PD-L1 within small EVs. EVs are small components secreted by cells involved in cellular communication. Much like normal cells, cancer cells are able to produce EVs. EVs are not only the exact reflection of the initial cell, but they are also representative of the modifications that characterize the initial cell [144]. Thus, in many cancers, the overexpression of PD-L1 observed on the surface of tumor cells is also found in the EVs. In particular, PD-L1 found on the surface of tumor-derived EVs promotes tumor progression and migration, contributes to the suppression of anti-tumor immune responses and mediates resistance to immunotherapies by binding directly to blocking antibodies [145]. However, there is another crucial question in the context of MPN: if the presence of PD-L1 in EVs is proven, do these extracellular forms of PD-L1 play a role in the development or evolution of MPN? Especially, are EVs with a high expression of PD-L1 potential biomarkers for the diagnosis or prognosis of MPN, similar to what has been demonstrated with circulating exosomal-PD-L1 in lung cancer, breast cancer [146], melanoma [147,148], and head and neck cancer [149,150]? So far, the presence of PD-L1 in EVs [151] and their existence in hematologic malignancies and MPN has not been proved [152], but there is evidence that large amounts of EVs can be found in the bone marrow compartment [153].

To conclude, the exploration of PD-1/PD-L1 inhibition immunotherapy in MPN may be worthwhile, both in the *JAK2* and *CALR* mutated forms. A phase II clinical study is currently underway to estimate the utility of PD-1 blockade (pembrolizumab) in MPN patients with primary or secondary MF [154]. Similar to melanoma, RCC, NSCLC, or Hodgkin lymphoma [155], these checkpoint blockade therapies are expected to be beneficial.

MPN present many significant clinical challenges for clinicians, and though considerable progress has been made in particular with the JAK1 and JAK2 inhibitor ruxolitinib, more effective alternative therapeutic approaches are needed. The available data suggest that HSPs and PD-1/PD-L1 are an opportunity and a challenge for MPN treatment. Together, they play a crucial role in most cancers, making them highly relevant for further research. The fact that many solid cancers exhibit higher expression of PD-1/PD-L1 means that there is a strong rational to use PD-1/PD-L1-targeted inhibitors. Immunotherapy is a promising treatment that may improve survival for some patients. Despite the many unknowns, including dose, schedule, patient selection, combination strategies, safety, durability, and even efficacy, therapies based on HSP and PD-1/PD-L1 blockades hold promise for the treatment of patients diagnosed with various hematological malignancies, including MPN.

Whether these new therapeutic strategies improve MPN treatment in the future or not, research into HSPs and the PD-1/PD-L1 axis will provide a better understanding of MPN pathogenesis and pave the way for new therapeutic strategies for all types of hematologic malignancies. Having more treatment targets would also facilitate the development of combination therapies for MPN, including approaches with the JAK1/2 inhibitor ruxolitinib or further agents that remain to be investigated. 

## Figures and Tables

**Figure 1 cancers-12-02592-f001:**
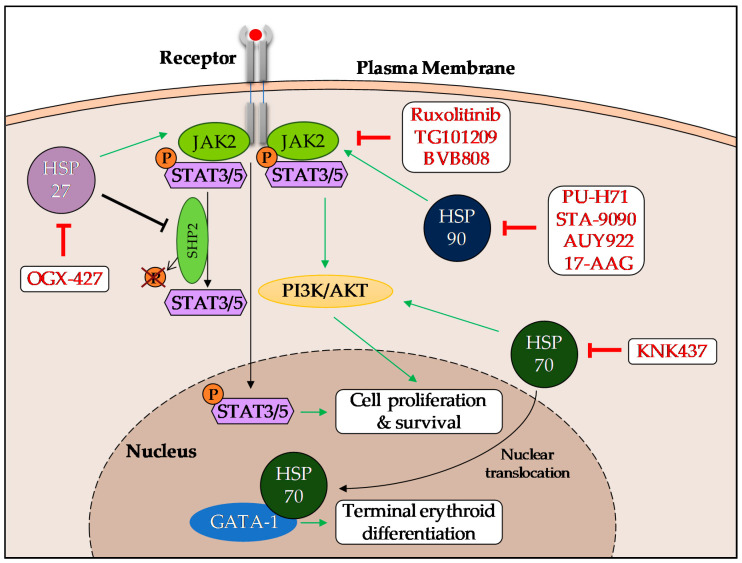
Role of HSPs and main known HSP interactions in MPN. HSP27 interacts with JAK2/STAT5 and prevents the SHP2-dependent de-phosphorylation of STAT5. HSP90 contributes to JAK2 stabilization and HSP70 activates the downstream target PI3K/AKT. Both HSP70 and HSP90 are also involved in inflammation, especially through an indirect regulation of NF-κB.

**Figure 2 cancers-12-02592-f002:**
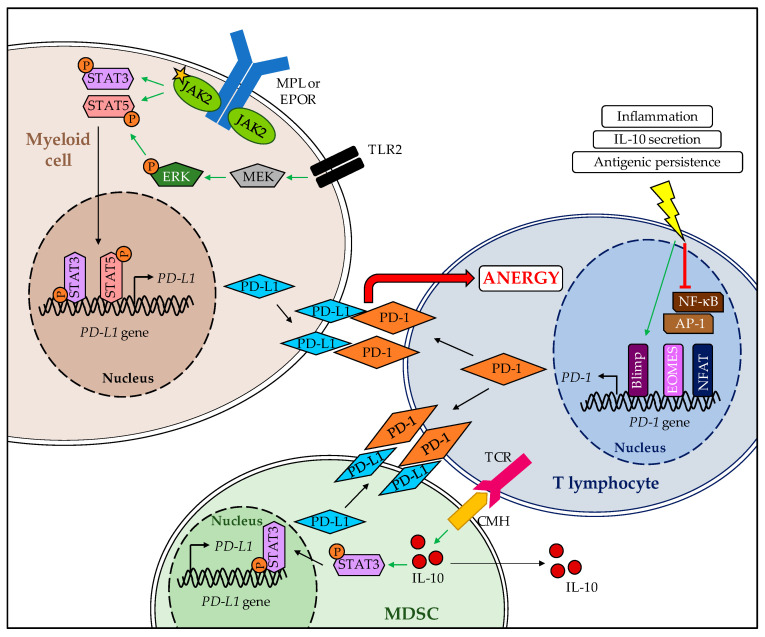
Role of PD-1/PD-L1 axis in MPN and its regulation. Constitutive activation of JAK2 receptor in myeloid cells leads to STAT-dependent expression of PD-L1, which then binds to PD-1 receptor when interacting with T lymphocytes triggering their exhaustion. In the meantime, TCR-CMH interaction in MDSC sets off IL-10 production and STAT3-mediated PD-L1 production. Finally, IL-10 production, constant inflammation and antigenic stimulation of T cells promote multiple transcription factor deeds (Blimp, EOMES, NFAT), leading to PD-1 expression and so reinforcing the phenomenon.

**Table 1 cancers-12-02592-t001:** Overview of the main strategies for HSP inhibition in MPN.

Inhibitor	Study Type	MPN Model	Ref.
Name	Nature	Structure
**Target: HSP90**
Ganetespib (STA-9090)	Synthetic, non-geldanamycin, small molecule inhibitor	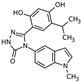	in vitro / preclinical	*JAK2V617F*-expressing cultured human (HEL92.1.7 and SET-2) MPN cells	[42]
17-AAG	Derivative of the antibiotic geldanamycin benzoquinone ansamycin	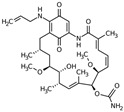	in vitro	*JAK2V617F*-expressing cultured human (HEL92.1.7 and UKE1) MPN cells	[41,45]
PU-H71	Non-ansamycin, purine scaffold inhibitor	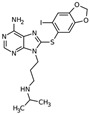	in vitro / preclinical	*JAK2V617F*-expressing cultured mouse (Ba/F3-JAK2-V617F) and human (HEL92.1.7 and UKE1), or primary human CD34+ MPN cells, *MplW515L*- and *Jak2V617F*- mouse retroviral bone marrow transplant MPN model	[40]
in vitro / preclinical	*Tp53*-KO/*Jak2V617F*- mouse retroviral bone marrow transplant model of post-MPN AML	[43]
preclinical	*MplW515L*- and *Jak2V617F*- mouse retroviral bone marrow transplant models of ET/MF	[44]
NVP_AUY922 (AUY922)	Esorcinylic isoxazole amide, 2nd generation non-geldanamycin inhibitor resorcinylic isoxazole amide	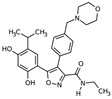	in vitro / preclinical	*JAK2V617F*-expressing cultured mouse (Ba/F3-JAK2V617F) and human (HEL92.1.7 and UKE1) or primary human CD34+ MPN cells, murine model of MF	[40,45]
clinical trial (phase II)	MPN	[52]
**Target: HSP70**
KNK437	Benzylidene lactam compound	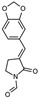	in vitro	*JAK2V617F*-positive cell lines and primary mononuclear and BM CD34+ cells from MPN patients	[61]
**Target: HSP27**
Apatorsen (OGX-427)	2nd generation 2′-methoxyethyl-modified ASOs	Sequence: 5′-GGGACGCGGC GCTCGGUCAU-3′	in vitro / preclinical	MPN-associated MF	[67]

AML: acute myeloid leukemia; ASOs: antisense oligonucleotides; BM: bone marrow; ET: essential thrombocytosis; MF: myelofibrosis; MPN: myeloproliferative neoplasm; PV: polycythemia vera; Marvin JS was used for drawing chemical structures, Marvin JS version 20.14, 2020, https://chemicalize.com/ developed by ChemAxon (http://www.chemaxon.com).

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
