# Peer review of "Heat Shock Proteins and PD-1/PD-L1 as Potential Therapeutic Targets in Myeloproliferative Neoplasms"

_cancers, 2020, doi:10.3390/cancers12092592_

Round 1

Reviewer 1 Report

Thank you for being given the opportunity to review the manuscript by De Almeida et al. The authors review the current state of the evidence regarding the role of HSPs and PD-1/PD-L1 in MPN. The manuscript is overall well-written and informative. Figures are excellent and lend support to the text body of the manuscript. However, I have the following comments/questions for the authors:

  • While I greatly appreciate that the author outline the pre-clinical evidence for various HSPs and PD-1/PD-L1 inhibitors, I think that especially the PD-1 chapter (chapter 4) is rather lengthy and not focused on myeloproliferative neoplasms as the title suggests. Several reviews have discussed PD-1/PD-L1 inhibition in detailed and referencing those should be sufficient. I would encourage the authors to focus more on MPNs.
  • Recently, Sorensen published an interesting report on PD-1 expression in myelofibrosis and PV showing that PD-1 and PD-L1 expression are increased compared to healthy donors and expression levels can be reduced by treatment with ruxolitinib (Eur J Haematol 2019 Oct;103(4):351-361. doi: 10.1111/ejh.13292.)
  • There have been several other interesting data on PD-1 in MPN recently that could be included (Holmstrom Oncoimmunology 2018; Holmstrom Leukemia. 2017;31(2):495–8.; Bozkus Cancer Discovery 2019)
  • There have been clinical trials of PD-1 inhibitors in MPN. Those should be included although results have not been published (NCT03065400, NCT02421354)
  • I would like to invite the authors to speculate on the potential to combine checkpoint inhibitors with JAK inhibitors or interferon.
  • It is a bit surprising to me that the authors discuss both HSPs and PD-1 in MPN in the same manuscript. Is there any rationale to just focus on those two new targets? Is there any data to suggest synergy between PD-1 and HSP inhibition? To me, it almost seems that those two themes run in parallel and the authors may want to consider including other targets as well vs focusing on only HSPs
  • If the authors decide to extend their discussion of the immunology of MPN the authors could discuss the anti-CD123 MoAb tagraxofusp, which has very promising results (EHA 2020)
  • JAK2 should be printed as italics when referring to the gene itself.

Reviewer 2 Report

In general this is an interesting review of potential treatments for MPN. My main question is why HSP inhibition and PD1/PDL1 exploration were chosen over other pathways as these two are completely unrelated. I've made minor comments but I would suggest editing throughout. Many sentences are wordy and difficult to read, and verb tense use is sometimes incorrect. 

Minor- in the introduction, authors only introduce ruxolitinib as a JAK inhibitor, but then refer to “treatments” plural, either change to singular or also include fedratinib, which also has approval for MF.

In the introduction, when introducing PD1 and PDL1, would recommend deleting “best known as” and just having (PD1) or (PD-L1).

Line 88-92 – The sentence that says HSPs are associated with poor prognosis is unclear. What about HSPs is associated with poor prognosis? Is it a mutation? Dysregulation?

Line 244-5- “block both HSP70 etc” – should say blocking

In general- the word interesting/interestingly is used often, would suggest eliminating many of these. 

Section on expression of PD-1 line 288- not clear what is meant by PDL1 being “carried” by a wide array of immune cells. Previously had said PD1 is expressed by t-cells etc, please clarify which cells express PD-L1

Minor grammatical errors are present in manuscript, would be careful with using same verbe tense throughout. Line 340- “with” should be deleted. Line 344-5 is awkward and should be clarified.

351- please clarify which cells express PD-1 and which express PD-L1

369-370- that line needs to be clarified starting with Although, monocytes and ending with ref 129. In general section 5.2 should be carefuly edited as a lot of the sentences are difficult to follow.

HSP sections- the authors mention the AUY922 study, however, do not mention the tolerability of PUH71, or much about that study.

427- the pembrolizumab study is not for ET and PV patients, but for MF patients with secondary MF or primary MF.

The introduction of EVs in the discussion is confusing for the PD-1 section. The authors should more clearly introduce the concept of EVs and how these relate to PD-L1 – Line 423 through 428. In particular 424 is confusing- to conclude… I am not sure what mutational burden is being referred to and how this relates to PD-L1.

Round 2

Reviewer 1 Report

The authors have addressed all my comments and I would recommend the manuscript in its current form for publication.

Reviewer 2 Report

I have read through the manuscript and believe the authors have addressed the comments of the initial submission.